# Aquarium Visitors Catch Some Rays: Rays Are More Active in the Presence of More Visitors

**DOI:** 10.3390/ani13223526

**Published:** 2023-11-15

**Authors:** Jordyn Truax, Jennifer Vonk, Eness Meri, Sandra M. Troxell-Smith

**Affiliations:** 1Department of Psychology, Oakland University, Rochester, MI 48309, USA; jptruax@oakland.edu (J.T.); vonk@oakland.edu (J.V.); 2Department of Biological Sciences, Oakland University, Rochester, MI 48309, USA; emeri55555@gmail.com

**Keywords:** zoo animal welfare, aquarium, elasmobranch, visitor effect

## Abstract

**Simple Summary:**

Research on the welfare of elasmobranchs (i.e., rays, sharks, and skates) in human care is lacking. Therefore, we observed the behaviors of members of four ray species in the presence of aquarium visitors. The rays spent more of their time performing active behaviors (i.e., swimming and eating) and less time performing inactive behaviors (i.e., resting and hiding) when there were more compared to fewer visitors present. However, there were no significant differences in other behavioral categories (i.e., negative and social) as a function of visitor number. To understand whether these findings applied to individual rays, we assessed the frequency of active versus inactive behaviors, and found that the three fiddler rays and one of the four southern stingrays’ active behaviors differed across visitor density. However, we did not find any significant associations between active behaviors and visitor numbers for the other three southern stingrays, the Atlantic stingray, or the blue-spotted maskrays. This study provides novel data on the activity budgets of four understudied species of rays. Although rays spent more time swimming in the presence of more visitors, future work is needed to determine whether these behavioral changes represent a positive or negative response to visitors.

**Abstract:**

Humans are a constant in the lives of captive animals, but the effects of human–animal interactions vary. Research on the welfare impacts of human–animal interactions focus predominantly on mammals, whereas fish have been overlooked. To address this lack of research, we assessed the impacts of aquarium visitors on the behaviors of ten members of four elasmobranch species: an Atlantic stingray (*Dasyatis sabina*), four southern stingrays (*Hypanus americanus*), two blue-spotted maskrays (*Neotrygon kuhlii*), and three fiddler rays (*Trygonorrhina dumerilii*). The rays engaged in a significantly higher proportion of active behaviors and a lower proportion of inactive behaviors when visitor density levels were high; however, there were no significant changes for negative or social behaviors. Individual analyses indicated that all three fiddler rays and one of the southern stingrays’ active behaviors differed across visitor density levels, whereas there was no association between active behavior and visitor density levels for the other rays. Further research is needed to determine whether this pattern is an adaptive or maladaptive response to visitors, but this research provides much needed initial data on activity budgets within elasmobranch species.

## 1. Introduction

One constant in the lives of captive animals is some level of human interaction, although the extent of these interactions varies according to how and where animals are housed [1,2]. These interactions can be quick and short-lived or more extensive, such as in environments where both caregivers and unfamiliar human visitors have the opportunity to directly interact with the animals (e.g., petting farms and touch pools). Even in such settings, human interactions are diverse and may include an animal caregiver delivering food, a trainer engaging in direct interaction, a veterinarian conducting an invasive exam, a child tapping on the habitat’s glass, or a group of visitors yelling in front of an open-air habitat [3,4,5]. As the nature of the interactions vary, so do the effects on the animals. Some human–animal interactions result in positive outcomes [6], whereas others may have neutral [7,8] or negative effects [9] for the animals. Given the omnipresence of humans in the captive environment, human–animal interactions have been incorporated into the Five Domains Model of animal welfare assessment [10]. This dominant model for guiding animal welfare assessments also includes the domains of nutrition, physical environment, health, and mental state. Although this model has been widely applied [11,12,13,14], there is little work assessing the welfare of aquatic species such as elasmobranchs (i.e., skates, sharks, rays and sawfish). We aim to remedy this omission by examining the behavior of captive rays in the presence of varying numbers of aquarium visitors.

Mammals have consistently been the primary focus of zoo and aquarium welfare research, whereas other animal classes, such as reptiles and fish, have been largely overlooked [15,16,17]. In fact, a review of animal welfare studies conducted in zoos and aquariums between 2008 and 2017, found that, of 310 articles, 74.8% (232/310) focused on mammals, 8.7% (27/310) focused on birds, 2.6% (8/310) on reptiles, 1.3% (4/310) on amphibians, and only 0.6% (2/310) of articles focused on fish [16]. Even with most research focusing on mammals, there is a bias toward particular species such as primates and felids [18]. A recent review of visitor impacts on non-primate species in zoos estimated that 56% of the study species were mammals, 28% birds, 9% reptiles, 4% fish, 2% amphibians, and 1% invertebrates [19]. Considering the popularity of aquariums, the underrepresentation of fish in animal welfare research is particularly concerning. The situation is further complicated by using the term “fish” to describe study subjects. Although scientifically accurate, the term “fish” can include vastly different species from two distinct taxonomic classes: bony fish (*Osteichthyes*) and cartilaginous fish (*Chondrichthyes*), of which elasmobranchs (skates, rays, and sharks) are one subclass [20]. Rose et al. [17] highlighted this disparity when they utilized a Web of Science search to investigate the number of zoo-focused research articles published between 2009 and 2018, and discovered that, of the 1434 articles analyzed, only 12 articles included “fish”, and only four of those articles specifically focused on Chondrichthyes, all of which were shark species.

Even in more commonly studied species, animal welfare is difficult to assess as complete accounts require a triangulation of methods. For example, fecal glucocorticoids can indicate arousal, but it is difficult to know if this arousal is adaptive or maladaptive without additional information, such as the environmental context [21]. Hormone analysis is most informative when paired with other indicators of health or behavioral assessments. Behavioral assessments are frequently used in captive settings due to their ease of collection, non-invasive nature, and ability to indicate changes from baseline behavior, such as increases in self-injurious behaviors or decreased time spent grooming. Although a multi-pronged approach through multiple assessment methods is optimal, in cases where that is not feasible, behavioral methods are preferred as they typically do not require expensive equipment or training and the results are more interpretable relative to other methods. For example, cognitive bias tests, a cognitive assessment to assess “optimism” versus “pessimism”, often require time-consuming training (see [22] for its use in zoos) and the results can be difficult to interpret [23]. In addition, behavioral assessments with understudied species are a valuable first step to exploring their welfare as it is necessary to establish baseline behavior before one can assess whether behavior is deviating in maladaptive ways in particular contexts.

Context is extremely important to determining the nature of the interaction between humans and animals and its potential effects on the animals. Interactions with keepers are typically beneficial to the animals, e.g., training sessions [24], although not always [25], whereas interactions with visitors are more complicated [1,18,19]. However, keepers may draw the attention of visitors when in animal enclosures, and visitor presence may distract the animal from the keeper, further complicating this issue [26]. The size [27], noise [28], and activity level of visitor groups [29] can all influence animal welfare, and negative behaviors are not uncommon in captive environments [3]. Some animals are more likely to be the target of these behaviors, particularly those perceived as more charismatic or those that are more active and spend more time closer to visitors [3,29]. The ability of animals to remove themselves from visitor presence through foliage or off-exhibit areas, which may also provide a refuge from noise [30], may mitigate potential deleterious effects [18]. Allowing opportunities to escape gives animals the ability to choose whether they want to experience visitors, and choice is an essential factor for positive welfare, even when animals do not utilize all options [31,32]. However, not all captive animals, including many aquarium animals, have access to off-exhibit areas, so examining the effects of visitors and accompanying noise on such animals is paramount. The presence of conspecifics may also help buffer the impact of high visitor density on social animals. For example, the size of meerkat groups appeared to moderate impacts of zoo visitors such that meerkats in smaller groups exhibited more stress behaviors under conditions of higher visitor density [9]. Environmental factors, which can vary along with visitor presence, such as temperature, humidity, or time of day, are necessary to consider when determining if changes in animal behavior are due to visitors or other confounding factors [26,33,34]. Thus, visitor effects are not independent of other contextual factors, and it is important to examine the animals’ use of escape behaviors and their social interactions to estimate their ability to cope with visitors.

Although extremely understudied, the limited research on elasmobranchs indicates that they may be impacted by visitor presence. For example, Lawrence et al. [35] assessed behavioral changes in two elasmobranch species (southern fiddler ray, *Trygonorrhina dumerilii*, and Port Jackson shark, *Heterodontus portusjacksoni*) following a habitat renovation that involved raising the viewing glass to prevent visitor interaction, and efforts to make the habitat more naturalistic. Following the renovation, the sharks displayed less abnormal behavior and the southern fiddler increased time spent resting, resulting in increased welfare outcomes for both species. Boyle et al. [36] also assessed the direct impact of visitors on the welfare of a variety of elasmobranchs, including cownose rays (*Rhinoptera bonasus*), southern stingrays (*Hypanus americanus*), bonnethead sharks (*Sphyrna tiburo*), brownbanded bamboo sharks (*Chiloscyllium punctatum*), and white-spotted bamboo sharks (*Chiloscyllium plagiosum*) housed in a touch pool. Both ray species and the bonnethead sharks exhibited behavioral changes in the presence of visitors. The bonnethead sharks moved more to the periphery of the touch pool when visitor food provisioning increased. The cownose rays became more solitary as the number of visitors increased, whereas the southern stingrays became more active when visitors would add food to their touch pool [36]. These results indicate that understanding visitor impacts on aquatic species is important, but also highlights that individual elasmobranchs vary in their responses to visitors. Furthermore, in this study, the provision of food was confounded with visitor presence, highlighting the need for further research.

We assessed the impact of differing numbers of visitors on four ray species (Atlantic stingrays (*Dasyatis sabina*), southern stingrays (*Hypanus americanus*), blue-spotted maskrays (*Neotrygon kuhlii*), and fiddler rays (*Trygonorrhina fasciata*)) housed in a non-touch tank at a midwestern aquarium. These are all benthic ray species that spend much of their time near or on the bottom of the ocean floor [37,38,39,40]. However, other aspects of their natural history vary substantially. For example, Atlantic stingrays have been observed to follow a more crepuscular activity pattern, thought to be affected by their nocturnal predators and crepuscular prey [38]. Southern stingrays are solitary foragers and naturally nocturnal but have been found to adjust their activity in response to provisioned food [41,42]. Blue-spotted maskrays are known to be relatively sedentary [39], whereas fiddler rays are relatively active during the day [43]. Thus, we anticipated some species differences in activity levels during the time of our observations.

To understand the influence of human visitors on ray behavior, we utilized focal instantaneous sampling to record the behavior of an individual ray, as well as the current number of visitors, using interval sampling. To assess a wide variety of behaviors, we created an ethogram based on previous behavioral research in rays (see Table 1) [44,45], including naturalistic behaviors seen in wild individuals, such as swimming, resting, hiding, and social interactions, as well as behaviors that may indicate negative welfare. Although there is not a wealth of research in this area for these ray species, we included abnormal swimming (i.e., swimming sideways) and wall flapping (i.e., flapping their fins at the surface of the water) in our ethogram, as hugging the wall can be detrimental to some elasmobranch species [46] and breaking the surface of the water has been viewed as a stereotypy or abnormal behavior in elasmobranchs [35,47]. As group factors have been shown to influence stereotypies in elasmobranchs [47], we also included interactions between the rays in our assessment. In addition, visitors have been shown to impact group dynamics in a variety of species, such as impacting the frequency of aggressive behavior in gorillas and lemurs [48,49] or affecting social proximity in kangaroos [27]. Thus, we hypothesized that the rays would engage in more positive social interactions during periods with lower visitor numbers and engage in more abnormal swimming behaviors and negative social interactions during periods with increased visitor numbers if they found high visitor densities stressful.

## 2. Materials and Methods

As these were observational studies, review of these studies was waived by the lead author’s institutional IACUC. Permission to conduct the study was granted by SeaLife Michigan Aquarium staff.

### 2.1. Subjects

We observed the behavior of 10 rays of four species: one female Atlantic stingray (*Dasyatis sabina*), four male southern stingrays (*Dasyatis americana*), two male blue-spotted maskrays (*Neotrygon kuhlii*), and three fiddler rays (*Trygonorrhina fasciata*; 2 males, 1 female). One male southern ray (#3) was removed from the tank for several weeks during data collection due to conspecific aggression; thus, there are fewer data points for this individual.

### 2.2. Habitat

All subjects were housed within the Stingray Bay Exhibit irregular nonagonal tank at Sea Life Michigan Aquarium, Auburn Hills, MI, USA. This aquarium is located within the Great Lakes Crossing Outlets Mall; thus, the aquarium is in a populous public area. The aquarium is open to the public every day from 10 a.m. to 6 p.m. (7 p.m. on Saturdays), except for certain holidays. The path through the aquarium travels through each exhibit, so visitors must travel through the Stingray Bay Exhibit to exit the aquarium. The tank had a volume of 20,500 L and a depth of approximately 1 m (39 in). The tank was kept on a 14:10 photoperiod, and water quality was held at 30–32 ppt salinity, with dissolved oxygen of 98–102%, pH of 7.7–8.0, and temperature of 23–24 °C. The habitat had two major fixtures: a large cylinder (seen as the tunnel in Figure 1) and a large stone archway (seen as the archway in Figure 1), along with a sandy substrate along the bottom of the tank. In addition to the focal rays, the tank also housed several white spotted bamboo sharks (*Chiloscyllium plagiosum*), a green moray eel (*Gymnothorax funebris*), and approximately 57 teleost fish of various species. The green moray eel generally remained within the large tunnel in the habitat, preventing its use by many of the other fish. Visitors were able to walk around and have visual access to almost all areas of the tank, except for a rocky outcropping toward the back of the tank where visitor viewing was limited to two viewing windows within a walkable tunnel (see Figure 1). Despite visitors being able to walk up to and look over the edges of the tank, touching the animals was not allowed.

### 2.3. Materials and Procedure

All data were collected via the ZooMonitor application [50]. Two observers recorded data throughout the study and did not start formal data collection until they obtained an inter-rater agreement of 80% or above for recording behavior within each observation interval for each individual ray across a data collection period. Inter-rater agreement sessions also served to ensure observers could reliably differentiate all ten rays. Data collection occurred from January to March 2023. A total of 6383 behavioral observations were recorded during the project period.

Observers recorded approximately 4–6 data collection periods throughout a given week. Observers recorded data throughout the week, except for Tuesday, at various times during the aquarium’s open hours: 10 a.m. to 6 p.m. EST on weekdays and Sundays and 10 a.m. to 7 p.m. EST on Saturday. We attempted to gain balanced observations across visitor numbers by conducting weekday and weekend observations. Observers also attempted to avoid observing during feeding times as the rays were seen to become more active around the presentation of food during inter-rater agreement sessions. Observers used instantaneous sampling with 30-s intervals, and once a behavior recording began, each individual ray was followed for ten minutes (20 total observations); therefore, each data collection period would last 90 or 100 min depending on whether southern ray #3 was present. We chose 30-s intervals as behaviors could change rapidly and we were especially concerned with capturing potentially rare behaviors (e.g., abnormal swimming, wall flapping). Observers used a random number generator to decide the order of focal individual follows for each observation period. At the start of each 30-s interval, observers recorded three pieces of information: the approximate number of visitors within the room, the subject’s location within the tank (see Figure 1), and one of the following behaviors: swimming, eating, resting, hiding, wall flapping, abnormal swimming, aggressive interaction, submissive interaction, displacement, neutral interaction, ray rest, not visible, or other (refer to ethogram for definitions). Behaviors were mutually exclusive, so only one was selected at each interval. The categories for the approximate number of visitors were decided based on the size of the room of the tank, observations of typical visitor numbers, and the estimated maximum number of individuals that could be in the room at one time. Visitor density was recorded as one of four categories: low (1–4 visitors), medium (5–9 visitors), high (10–14 visitors), or very high (15 or more visitors).

### 2.4. Statistical Analyses

Analyses were conducted in R version 4.2.1 [51] for the multilevel model and IBM SPSS v. 26 for all chi-square tests. To ensure we analyzed only intervals involving known behaviors, we removed all intervals in which the subjects were “not visible” from analyses. For the purpose of analyses, we grouped behaviors into four categories: active, inactive, negative, and social (see Table 1). There were no instances of the coders selecting “other”, so this category was not included in analyses.

We conducted four multilevel models to control for the dependence due to individual, and potentially due to species. As we had a limited sample across species and individuals, we did not include any individual- or species-level predictors. Species and individuals were included in the model only to control for potential variance due to these factors. The nlme package in R was used for these analyses [52]. A multilevel model was conducted for each behavioral category (active, inactive, negative, and social) with a binomial outcome variable (present/not present) and visitor density (low, medium, high, very high) used as our predictor variable. The models also included a random intercept for individual and dummy coded variables for each species except our reference species (Atlantic stingray).

Similar to previous research [36,53,54], to investigate whether the significant group-level effects were representative of the individual, we also conducted multiple chi-square tests for each individual ray to assess whether the distribution of each ray’s behavior across categories differed between visitor numbers. Negative and social behaviors did not occur very frequently, which would have violated one of the assumptions of the chi-square test with regard to very low expected cell values; thus, we combined behaviors into two categories. As the proportions of active and inactive behaviors were found to vary significantly across visitor levels across all rays, we compared active behaviors with a combined category of not active behaviors, which included inactive, negative, and social. Thus, these chi-square tests assessed whether the frequency of active behaviors versus the remaining other categories of behavior differed across visitor density levels.

## 3. Results

Our multilevel models indicated that visitor density predicted the presence of active behaviors (*Z* = 4.429, *p* < 0.001) and inactive behaviors (*Z* = −4.460, *p* < 0.001), but did not predict the presence of social (*Z* = −1.755, *p* = 0.079) or negative behaviors (*Z* = 0.576, *p* = 0.564). Active behaviors were 2.199% more likely to occur across each increase in level of visitor density, whereas inactive behaviors were 2.091% less likely to occur across each increase in level of visitor density. Figure 2 shows the changes in the proportion of observed behaviors across categories as visitor density increased.

Individual chi-square tests revealed a significant association between active or not active behavior and visitor density level for all three fiddler rays (fiddler 1: χ^2^ = 16.392, *p* < 0.001; fiddler 2: χ^2^ = 10.274, *p* = 0.016; fiddler 3: χ^2^ = 27.306, *p* < 0.001), as well as a trend toward significance for southern stingray 3 (χ^2^ = 8.486, *p* = 0.037) and southern stingray 2 (χ^2^ = 7.401, *p* = 0.060. There was no association between active or not active behavior and visitor density level for the Atlantic stingray (χ^2^ = 1.771, *p* = 0.621), both blue-spotted maskrays (blue-spotted 1: χ^2^ = 1.719, *p* = 0.633; blue-spotted 2: χ^2^ = 2.301, *p* = 0.512), or the other two southern stingrays (southern stingray 1: χ^2^ = 0.441, *p* = 0.932; southern stingray 4: χ^2^ = 0.448, *p* = 0.930). Figure 3 shows how the frequency of the four categories of behavior varied across visitor density levels for each individual ray.

## 4. Discussion

Our results indicate that the rays spent a larger proportion of their time engaged in active behaviors and a smaller proportion of their time engaged in inactive behaviors as visitor density increased. However, they did not display significant differences in social or negative behaviors across visitor density levels. When we further analyzed these differences by individual, we found that all three of the fiddler rays’, and one of the southern stingrays’ active behaviors significantly differed by visitor density levels, whereas this was not found for the other three southern stingrays, the Atlantic ray, and the blue-spotted maskrays.

Given the paucity of behavioral research on captive rays, our results contribute to a better understanding of the behavioral effects of visitors on these fish in captive settings. The observation of changes in activity budgets under different environmental conditions is a well-established method to identify potentially deleterious environmental conditions for animal well-being [35,55,56,57]. As rays are common species housed in aquariums with high densities of visitors and a high intensity of interactions (i.e., in touch pool exhibits), it is critical to examine whether they may be impacted by such interactions. It is encouraging that the present results indicate no significant increase in negative or aggressive behaviors and no significant decrease in social behaviors (which can potentially buffer against negative impacts of visitors, e.g., [9]). However, we did observe a significant increase in activity and a significant decrease in inactivity when visitor numbers were greater, which was driven by members of two of the observed species.

It is difficult to conclude whether this increase in activity represented a positive or negative impact of visitor density. Whereas the absence of an increase in conspicuous abnormal or aggressive behaviors is encouraging, the decline in inactive behaviors during periods of high visitor density may indicate a disruption of rest, and the increased swimming displayed by the rays may be indicative of increased stress. Lawrence et al. [35] reported that a southern fiddler ray (*Trygonorrhina dumerilii*) increased its resting behaviors in response to alterations to its habitat that decreased visitor viewability, which is in line with that species’ natural history. Many ray species spend a significant amount of their time resting on or burying themselves in the sand [58,59,60], which indicates that this is an important behavior for captive rays. In addition, Boyle et al. [36] found that southern stingrays increased their swimming behaviors and decreased their resting behaviors in response to visitors, but the visitors were provisioning food to the rays and could directly touch the rays, which was not the case in our study. A recent review of visitor effects on non-primate species indicated that zoo visitors were most likely to have a neutral or unknown impact on fish in general, although again, many of the studies assessed responses to interaction or handling [19]. These impactful differences make it difficult to directly compare our results and to determine whether the changes we observed indicate beneficial, neutral, or harmful effects of visitors.

Aside from the observed effects of visitor presence on the activity level of the rays, we also present useful data on general activity budgets in several ray species in a captive setting. Activity budgets are useful as they can indicate the extent to which captive animals behave similarly to their wild counterparts, which may be a positive indicator of welfare [57]. Most of these rays spent the majority of their time being inactive, which may reflect ideal natural behavior given that they were not observed often during feeding time and they were observed during the day (10 a.m. to potentially 7 p.m.). We also witnessed relatively few social interactions of either a positive or a negative nature, which may be important for determining how many rays should be socially housed and at what densities.

### Limitations and Future Directions

Our study was limited by a small number of individuals within each species, so we could not explore species differences. However, the results tentatively suggest larger effects of visitor presence on some species than others, which would have to be explored with larger samples. It is yet to be determined whether the individual rays in our study that showed changes in behavior across visitor density levels did so mainly due to individual factors (e.g., temperament or previous experience), species differences, or other unexpected factors. Given the previous literature on the influence of individual and species differences on the impact of visitors in non-fish species [18,49], it would be helpful to see this literature expanded to under-researched groups. We also had a relatively small number of observations, and we did not observe the rays when the aquarium was closed to the public (i.e., in the absence of visitors). As 24/7 approaches to welfare are gaining traction [61], it is important to know how their behaviors may change throughout the night, as well as during the day. Our study focused on solely the number of visitors present, rather than particular visitor factors, such as noise level and behavior. As loud and fast-moving visitors can be threatening to animals [18], and research has shown that fish are impacted by noise [62], future research on visitor impacts on rays would benefit from considering these factors. We also did not assess any contextual factors that may account for responses to zoo visitors, such as time of day [33]. However, the aquarium was indoors, which removed impacts of weather, and staff heavily regulated other contextual factors, such as temperature. Given the relatively small dimensions of the tank, the fact that some areas could not be accessed by the larger rays, and the lack of deep or off-exhibit escape areas, we were not able to meaningfully examine the rays’ use of space as a function of visitor density. In larger tanks with areas where rays can escape from view or contact (especially in touch pools), it would be important to examine whether they take advantage of opportunities to escape more frequently with more visitors surrounding their habitats. In deeper tanks, it would also be more feasible to explore the rays’ relative position to the surface or bottom of the tank.

However, as one of the very few studies of ray behavior as a function of visitor density, our results point to the importance of further study of these fascinating and commonly housed elasmobranchs. Additional research is certainly needed to definitively determine whether increased swimming activity and decreased resting is an adaptive or maladaptive response to increased aquarium visitor presence in elasmobranchs. However, such results are important to note, as welfare studies tend to focus on the presence of more conspicuous behaviors (i.e., abnormal/stereotypic) to indicate possible negative welfare [1,63,64], leaving the possibility that more subtle activity changes could be overlooked. As researchers continue the important task of considering and including more understudied species in captive welfare studies, activity budgets should be inclusive of a wide range of behaviors to meaningfully assess impacts on welfare. For rays specifically, future studies are needed to establish typical ray behavior in captive settings, and only once this is established can researchers and care staff better assess factors that negatively impact welfare.

## 5. Conclusions

Overall, this study provides essential information on activity budgets in rays. Our results suggest that visitors can impact the behaviors of rays, particularly the proportion of their time spent engaging in more active behaviors such as swimming or inactive behaviors such as resting, although not all of the rays showed behavioral differences at varying visitor levels. Future research may be able to further examine whether increased activity and decreased rest are reflective of positive or negative impacts on ray welfare and assess the importance of documenting individual behavioral differences along with differences across ray species.

## Figures and Tables

**Figure 1 animals-13-03526-f001:**
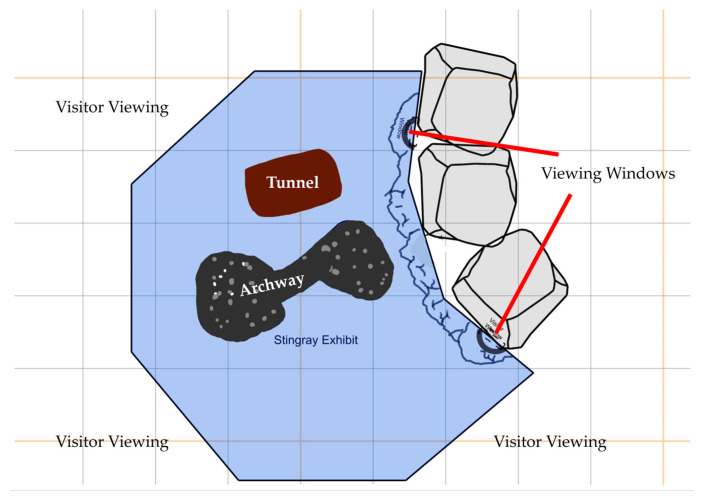
Diagram of stingray habitat observed on ZooMonitor.

**Figure 2 animals-13-03526-f002:**
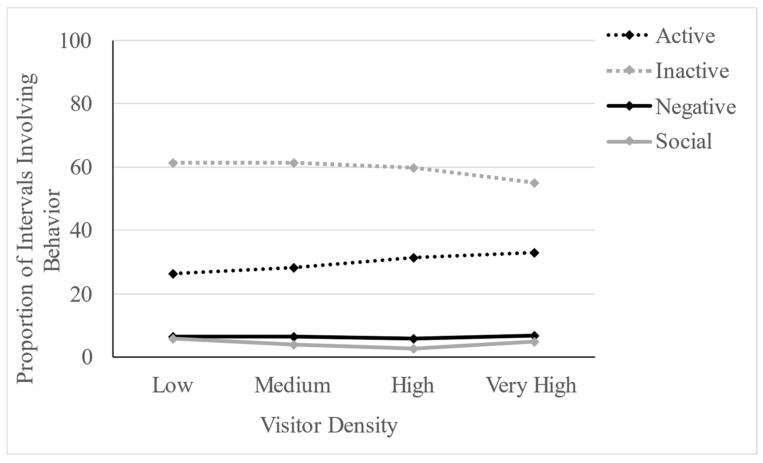
The proportion of observed intervals involving each behavior (active, inactive, negative, or social) across visitor density levels (low: 1–4, medium: 5–9, high: 10–14, very high: 15 or more). Dotted lines indicate *p* < 0.001, solid lines are non-significant.

**Figure 3 animals-13-03526-f003:**
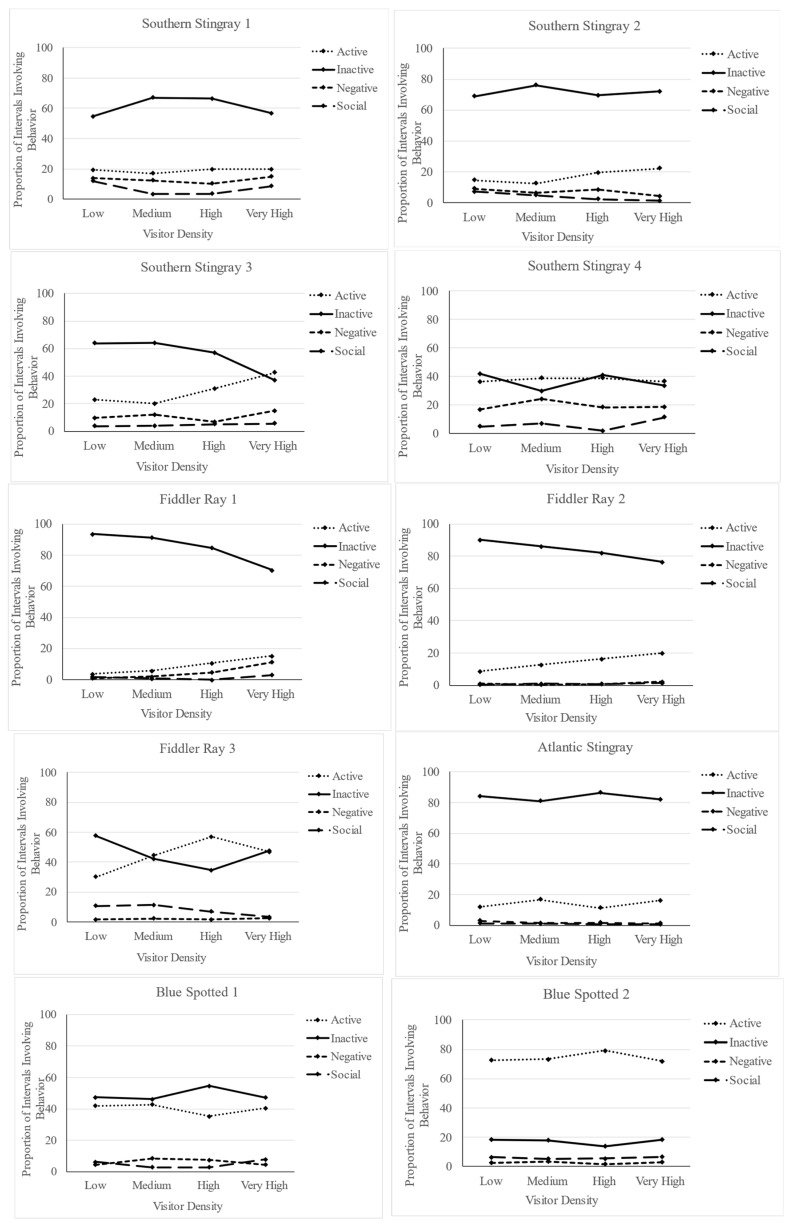
Proportion of observed intervals involving each behavior (active, inactive, negative, social) for each individual ray across visitor density levels (low: 1–4, medium: 5–9, high: 10–14, very high: 15 or more).

**Table 1 animals-13-03526-t001:** Ethogram of ray behaviors.

Category	Behavior	Description
Active	Swimming	Moving through the water without touching the tank wall
	Eating	Actively consuming food items
Inactive	Resting	Remaining stationary on the tank floor, the individual may be covered in sand or lying on top of the sand (but not on top of another individual)
	Hiding	Remaining stationary on the tank floor while covered in sand or in the process of covering themselves in sand
Negative	Wall Flapping	Moving fin(s) back and forth in a flapping motion while the body is partially out of the water and the ventral face is touching the tank wall
	Abnormal Swimming	Moving through the water in an atypical fashion, including vertically (i.e., while the ventral face is parallel with the tank wall) or upside down (i.e., while the ventral face is directed toward the water surface)
	Aggressive Interaction	Behaving in a hostile manner toward another ray, which includes biting, chasing (i.e., swimming behind another ray at a quick speed), nose shoving (i.e., pushing its snout into another individual), and tail raising (i.e., raising tail in a defensive manner while facing another ray)
	Submissive Interaction	Behaving in an unassertive manner toward another ray, which includes avoiding (i.e., turning away from another ray when that individual becomes visible) and giving way (i.e., an individual moving out of the path of another ray who continues on their current path)
	Displacement	Moving toward another ray causing this individual to move away
Social	Neutral Interaction	Behaving in a neutral manner toward another ray, which includes swimming over the top of another ray, following (i.e., swimming behind another ray at a slow speed), and parallel swimming (i.e., swimming next to another ray at the same speed).
	Ray Rest	Remaining stationary while on top of another ray
Non-classified	Not Visible	Out of view of the observer
	Other	The behavior cannot be classified into the above categories

## Data Availability

The data are available upon request from the corresponding author.

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
