# Peer review of "Aquarium Visitors Catch Some Rays: Rays Are More Active in the Presence of More Visitors"

_animals, 2023, doi:10.3390/ani13223526_

Round 1

Reviewer 1 Report

Comments and Suggestions for Authors

Overall the paper aims to bring new information into an understudied field. There are several contributions the paper brings to aquarium and welfare research. The use of  Zoomonitor for data collection allows for repeatability within other aquariums. The range of elasmobranchs utilised gives a good range for comparison and adds activity budget information in a novel species. Consideration of visitor numbers in relation to activity allows application to real world scenarios.

Article: The article gives consideration into replication and testability by using species that had multiple individuals in the aquarium. Within the methodology it is hard to determine, however the number of observations that were actually recorded. Authors mention when the sampling occurred but not the number of observations overall. Justification of the groups of high and low visitors would help potential comparisons with other visitor studies, particularly due to no other information being taken such as auditory disturbance.

Review:  A good review of the topic is given in the introduction. The discussion would benefit from linking existing literature to their findings. The discussion covers their findings but lacks further depth by relating it to the current field of research of visitor disturbance or effect. Questions also arise over whether the individual activity budget is needed or whether they can be combined.

Author Response

Dear reviewer,

We thank you for your comments on our paper. Please find our response attached. 

Best,

Jordyn Truax

Reviewer 2 Report

Comments and Suggestions for Authors

This is a very slight paper that reveals some small variations in swimming behaviour in an aquarium and links it to visitor numbers. The choice was made to divide visitor numbers into 2 groups lees than 10 and 10 and above because 'this categorisation best captured the variability in ray responses' which may be rephrased 'this categorisation gave us the result we hoped for'. You must, at least, present the data for all four visitor numbers.  If there was no suggestion of a difference between the <5 and >15 groups. this will blow your conclusions out of the water.

Fig 1 illustrates an irregular nonagon, not an octagon

Author Response

(The authors gave the same response as above.)

Reviewer 3 Report

Comments and Suggestions for Authors

The referencing format is incorrect for the journal.

It is best to avoid starting a sentence with "Because..." as the reader doesn't know what is being compared back to.

Maybe remove the scientific names from the simple summary and include them in the formal abstract?

Lines 39 to 46 are poorly referenced. You need citations to support some of your arguments and pieces of information here.

I see you have included the paper by Collins (on negative behaviours of zoo visitors and animal responses) in your work but it might be worth providing a deeper evaluation of why these visitor actions are negative. 

Also, you should consider this paper too: https://www.mdpi.com/2076-2615/13/16/2661 (again by Collins that adds further information on what causes negative visitor behaviours). 

The section on the visitor effect being complex needs to be better evaluated and discussed. For example, there are papers that have shown the difference in an animal's responses to visitors compared to keepers https://www.frontiersin.org/articles/10.3389/fvets.2020.00236

And that show that the impact of environmental variables as potentially more important that visitor presence https://onlinelibrary.wiley.com/doi/full/10.1002/zoo.21615 and https://www.jzar.org/jzar/article/view/343 

Your concept of "noise" should also be extended, especially in terms of how it could impact on aquatic animals. There are some useful papers on sound measurement, visitor effects and animal responses that you could consider evaluating. 

For example https://www.sciencedirect.com/science/article/pii/S0376635722001802 and https://doi.org/10.1002/ajp.23421 

Line 167: positive social interactions between rays? Or between rays and visitors?

Is there justification for the 30 second interval for behavioural recording? Do rays change their behaviours quickly and in a short amount of time? If so, the short recording point is fine. If not, are you not worried there may be pseudoreplication of data points, as the short recording point would mean that animal behaviour is not naturally changing between observations, and you are over-inflating the number of data points on the same behaviour?

The ethogram is good, although some of the mechanical descriptions of behaviours could be extended to provide further details on what they look like.

Is there any precedent or evidence for the visitor density categories? Not a criticism, might be helpful for others to have an explanation of how these values were judged for this style and type of exhibit at this specific type of zoo.

Also, visitor density is not, in itself, a type of behaviour so this is inappropriate to the ethogram and should be in a separate table.

There is no data analysis section in the method. Data analyses are incorrectly described at the start of the results. You need to explain your subjects, location, data collection techniques and data analyses in the Methods section. The Results is for the description of the outputs from your testing. 

You conducted one-way ANOVAs on these behavioural data. Were data tested for normality and equal variance before testing commenced? Were all fish individually identifiable? And how does this ANOVA cope with species differences?

Given you have repeated measures on the same animals, it is probably worth considering a re-run of this analyses using a repeated measures ANOVA (so you would include date as the random factor) and then you could also include individual variation in your model too. 

Figure 2: What do you mean on the Y axis "Behavior Proportion"? Does this need units?

There are elements of discussion in the Results section. You do not discuss what you found in this section, you simply state your findings and describe the outputs of your testing. Please remove all discursive sections to the Discussion. Likewise, there are elements of the Results that explain precedent for the analyses and testing, and this information would be placed in the Method (data analysis section), e.g. from line 266 onwards. Please edit throughout. 

Line 283: there is an extra s on spotted

Figure 3: On the X axis, proportion of what? These figures are quite small. Is there a better way to present all animals on one larger graph?

Please ensure you label all graph axes with useful information that shows the reader what data are presented on the graph.

Your discussion is useful, and you have highlighted the small sample size as a limitation. But if you consider this in your inferential analyses (i.e. consider the implications of repeated observations on the same population) you would have already tried to control for this in your experimental design. 

Please start the discussion with an overall summary of your key results outputs that you will then go on to evaluate and explain further. 

Line 326: perhaps "you present useful data on ray behaviour" rather than some of the first. How do you know this? 

Line 338: You haven't tested for species differences empirically and you have only a very small number of some species and no replication. I don't believe you can clearly state species differences unless you try and model for it or you have repeats on other individuals of that species. 

I think you should frame your discussion as a case study on how rays (as a whole) can respond to visitors but to be clear on any potential (descriptive) species differences in behaviour, replication is needed across populations as well as testing of personality and individual fish responses (i.e. some fish may cope better than others). 

Line 344: hopefully rays can also escape to deeper areas and away from visitors?

Comments on the Quality of English Language

Please check some sentence structure in some areas of the manuscript.

Author Response

(The authors gave the same response as above.)

Round 2

Reviewer 3 Report

Comments and Suggestions for Authors

The authors have made some useful amendments to the manuscript and I believe it is suitable for publication. I would recommend ethical review for all methods moving forward, irrespective of them being observational or not, as to have data collection procedures peer reviewed and considered by others does, in itself, make science more ethical. 

Comments on the Quality of English Language

Style of written English is very good.